# TNAP and P2X7R: New Plasma Biomarkers for Alzheimer’s Disease

**DOI:** 10.3390/ijms241310897

**Published:** 2023-06-30

**Authors:** Paloma Aivar, Carolina Bianchi, Caterina Di Lauro, Lucia Soria-Tobar, Beatriz Alvarez-Castelao, Miguel Calero, Miguel Medina, Miguel Diaz-Hernandez

**Affiliations:** 1Departamento de Bioquímica y Biología Molecular, Facultad de Veterinaria, Universidad Complutense de Madrid, 28040 Madrid, Spain; paloma.aivar@universidadeuropea.es (P.A.); cbianchi@ucm.es (C.B.); cdilauro@ucm.es (C.D.L.); lusoria@ucm.es (L.S.-T.); balvar03@ucm.es (B.A.-C.); 2Departamento de Ciencias de la Salud, Facultad de Ciencias Biomédicas y de la Salud, Universidad Europea de Madrid, 28670 Madrid, Spain; 3Instituto de Investigación Sanitaria del Hospital Clínico San Carlos, IdISSC, 28040 Madrid, Spain; 4Centro de Investigación Biomédica En Red-Enfermedades Neurodegenerativas (CIBERNED), 28029 Madrid, Spain; mcalero@fundacioncien.es (M.C.); mmedina@ciberned.es (M.M.); 5Alzheimer Disease Research Unit, CIEN Foundation, Queen Sofia Foundation Alzheimer Center, 28031 Madrid, Spain

**Keywords:** P2X7R, TNAP, biomarker, Alzheimer

## Abstract

Over the last few years, intense research efforts have been made to anticipate or improve the diagnosis of Alzheimer’s disease by detecting blood biomarkers. However, the most promising blood biomarkers identified to date have some limitations, most of them related to the techniques required for their detection. Hence, new blood biomarkers should be identified to improve the diagnosis of AD, better discriminate between AD and mild cognitive impairment (MCI) and identify cognitively unimpaired (CU) older individuals at risk for progression to AD. Our previous studies demonstrated that both the purinergic receptor P2X7 and the tissue-nonspecific alkaline phosphatase ectoenzyme (TNAP) are upregulated in the brains of AD patients. Since both proteins are also present in plasma, we investigated whether plasma P2X7R and TNAP are altered in MCI and AD patients and, if so, their potential role as AD biomarkers. We found that AD but not MCI patients present increased plasma P2X7R levels. Nevertheless, TNAP plasma activity was increased in MCI patients and decreased in the AD group. ROC curve analysis indicated that measuring both parameters has a reasonable discriminating capability to diagnose MCI and AD conditions. In addition to confirming that individuals progressing to MCI have increased TNAP activity in plasma, longitudinal studies also revealed that CU individuals have lower plasma TNAP activity than stable controls. Thus, we propose that P2X7 and TNAP could serve as new plasma biomarkers for MCI and AD.

## 1. Introduction

Alzheimer’s disease (AD) is the most common form of dementia, potentially contributing to 60–70% of cases [1]. Until recently, AD patients were diagnosed based solely on clinical symptomatology, with a definitive neuropathological confirmation obtained postmortem by identifying the presence of two histopathological hallmarks associated with this disease, neurofibrillary tangles (NFTs) of hyperphosphorylated tau protein and senile plaques formed by beta-amyloid (Aβ) protein [2,3]. The AD diagnostic criteria improved once technical advances allowed both hallmarks to be measured in vivo by using cerebrospinal fluid (CSF) samples or magnetic positron emission tomography (PET) [4,5,6]. However, both the invasiveness associated with CSF sample collection and the elevated cost of PET limit the widespread use of both techniques in the primary care setting, where most AD diagnoses are made [7]. Current efforts are directed toward the identification of blood biomarkers, because blood samples are more accessible than CSF samples, and their associated analysis costs are lower than those of PET imaging [8]. The most promising blood biomarkers identified to date include the Aβ42/Aβ40 ratio [9], phosphorylated tau [10,11] and neurofilament light (NfL) [12]. Despite the promising role of plasma Aβ as an AD biomarker, the single-molecule array (Simoa) or immunoprecipitation mass spectrometry (IP/MS) techniques required for its detection are costly and require extensive development before being used in the primary care setting [8]. Moreover, plasma Aβ42/Aβ40 levels may be affected by preanalytical handling and analytical performance [13]. These limitations strongly suggest that additional efforts should be made to identify new blood biomarkers, not only to improve the diagnosis of AD but also to allow better discrimination between mild cognitive impairment (MCI) and AD patients. In addition, improvements must be pursued toward the identification of cognitively unimpaired (CU) older individuals at risk for progression to AD and the monitoring of the effects of disease-modifying therapies in clinical trials [14].

Over recent years, several pieces of evidence have accumulated suggesting that the purinergic receptor P2X7 (P2X7R), an ATP-gated ionotropic receptor, is involved in AD-associated pathology [15]. In fact, it was reported that P2X7R regulates senile plaque formation [16], tau-induced cell death [17,18] and amyloid peptide- or tau-induced neuroinflammation [19]. Interestingly, tissue-nonspecific alkaline phosphatase (TNAP), the brain-expressed alkaline phosphatase (AP) isozyme, regulates the ATP concentration in the close environment of the P2X7 receptor and modulates its function [20]. TNAP has also been related to AD-associated pathology [21]. This is due to the fact that TNAP can dephosphorylate a broad range of extracellular substrates, including the extracellular hyperphosphorylated tau protein [22], and thus impact tau-induced toxicity spreading [23]. Supporting the involvement of both elements, P2X7R and TNAP, in AD-associated pathology, it was also reported that (i) AD patients and mouse models mimicking this disease present both proteins with altered expression levels and function [17,18,23], and (ii) the genetic ablation or selective pharmacological blockage of each of these targets reverts AD-associated behavioral deficits and elongates the life expectancy of different mouse models mimicking this disease [16,17,18,23]. Nevertheless, the possible role of these proteins, either individually or jointly, as plasma AD biomarkers has not been studied yet. While there have been preliminary works pointing to AD patients showing increased alkaline phosphatase activity in plasma [24,25], they did not provide any evidence indicating whether this alteration also takes place in MCI patients or in CU individuals. On the other hand, recent studies have reported that patients with neural disorders causing neuroinflammation, such as those diagnosed with temporal lobe epilepsy, present elevated plasma levels of a soluble P2X7R form [26,27]. This soluble receptor form would come from the activation of P2X7R, since this also leads to the cellular shedding of this receptor [28]. In the present work, we assayed the plasma P2X7R levels and specific TNAP plasma activity in a slight but well-established cohort of 60 persons, including individuals diagnosed with AD or MCI and CU or non-affected individuals.

## 2. Results

### 2.1. Study Population Characteristics

A total of 60 individuals were included in the present study, split into the following groups: 23 non-affected control subjects, 17 patients with mild cognitive impairment (MCI) and 20 patients with Alzheimer’s disease (AD), all of them well matched for gender and age (Table 1). The classification of patients into the control, MCI or AD group was based on neuropsychological assessment and white matter hyperintensities (WMHs) measured on the Fazekas scale (FS) (as described in the Methods section). According to the FS, 41% of MCI patients and 50% of AD patients were in stage 2 (pathological brain with early confluent WMH) or 3 (pathological brain with diffuse confluent WMH) in comparison to 13% of control patients. The neuropsychological assessment included, among others, the Mini-Mental State Examination (MMSE), which measures global cognitive function, the Functional Activities Questionnaire (FAQ) and the Clock Drawing Test (CDT), which focuses on functional decline. As expected, most AD patients were not able to complete the neuropsychological assessment. In MCI patients, cognitive test results reflect the beginning of dementia (Table 1).

For MCI patients, the MMSET score grew closer to the 24 score threshold for cognitive decline (MMSETMCI = 27.2 ± 1.7; MMSETControl = 29.3 ± 0.7, Table 1), the FAQ was significantly increased in comparison to the control group (FAQMCI = 3.4 ± 1.5; FAQControl = 0.5 ± 0.9; the threshold is 6, Table 1), and the CDT performance declined (CDTMCI = 8.5 ± 1.7; CDTControl = 9.7 ± 0.6, Table 1). As depression is an important risk factor for dementia, especially for women, we confirmed that both control and MCI patients had a good global depression score, below 5 (GDSMCI = 1.8 ± 2.2; GDSControl = 1.0 ± 1.4, threshold 5, Table 1). Finally, based on the neuropsychological assessment results, MCI patients received a Clinical Dementia Rating of 0.5, which corresponds to the initial state of dementia (Table 1).

Since the *APOƐ4* genetic variant has been established as the strongest genetic risk factor for late-onset AD [29], we calculated the proportion of subjects with at least one copy of the *ApoƐ4* allele in all groups. It is noted that there was a greater proportion of subjects with at least one copy of the *ApoƐ4* allele in both MCI and AD groups (control = 8.7%; MCI = 23.5%; AD = 55%, Table 1).

### 2.2. P2X7R Plasma Levels Are Increased in AD but Not in MCI Patients

Plasma P2X7R levels were measured by ELISA, as described in the Methods section. Our analysis revealed that non-affected individuals presented plasma P2X7R levels of 37.4 ± 3.1 pg/mL (Figure 1A), a value that is within the range reported by other groups (16.7–82.1 pg/mL) [28]. No difference in P2X7R plasma levels was noticed in MCI patients in comparison with the control group (41.3 ± 3.1 pg/mL). However, AD patients presented a significant increase in plasma P2X7R levels (75.5 ± 12.9 pg/mL) when compared to those detected in control or MCI groups (Figure 1A).

### 2.3. TNAP Activity Is Increased in MCI but Not in AD Patients

Since previous works have reported a close relationship between P2X7R and TNAP [20,30], in parallel, we also measured TNAP activity in the plasma of the same cohort of patients. The results show that TNAP activity was significantly increased in MCI patients in comparison with non-affected controls (an increase of 31.7 ± 8.4% compared to controls, with a mean TNAP activity in controls of 51.3 ± 3.6 UI/L, Figure 1B). On the contrary, a non-significant reduction in TNAP activity was seen in AD patients when compared to controls (a reduction of 14.2 ± 6.3% compared to controls, Figure 1B). In good agreement, when TNAP activity levels were compared between MCI and AD patients, a significant decrease was noticed (a reduction of 36.0 ± 9.6% compared to MCI, Figure 1B). These results suggest that TNAP activity may present biphasic behavior with the progression of the disease.

We then represented both P2X7R and TNAP activity for every individual patient and found that patients with the same condition were grouped into point clouds with significantly different centers of mass (Figure 1C). The correlation between the MRI Fazekas scale and P2X7R levels or TNAP activity in all diagnostic groups was non-statistically significant (see Appendix A). Along the same lines, we performed a correlation analysis between all neuropsychological assessments and both parameters for the control and MCI groups (see Appendix A). P2X7R levels only reached a significant positive correlation with MMSE in the MCI group. Similarly, only in the MCI group was a significant positive correlation obtained between TNAP activity and the Global Depression Scale, consistent with recently published data [31].

### 2.4. Measuring Both P2X7R and TNAP Activity Has a Reasonable Discrimination Capability to Diagnose MCI and AD Conditions

Receiver operating characteristic (ROC) analysis was performed to investigate the diagnostic potential of measuring plasma P2X7R protein and TNAP activity for diagnosing AD. Since ROC curves depend on the patient’s characteristics and the disease spectrum, we tested separately for MCI and AD conditions. Individual receiver operating curve (ROC) analysis revealed that the measurement of P2X7R plasma levels was not useful for differentiating between control and MCI patients (AUC of 0.65 ± 0.09, sensitivity 78.6% and specificity 50.0%, Figure 2A, Table 2); however, it had acceptable accuracy for discriminating between control and AD patients (AUC of 0.73 ± 0.08, sensitivity 90.0% and specificity 50.22% Figure 2B, and Table 2). Regarding TNAP plasma activity, this parameter showed better diagnostic accuracy in identifying MCI patients (AUC of 0.75 ± 0.09, sensitivity 71.4% and specificity 72.3%, Table 2) than AD patients (AUC value 0.69 ± 0.08, sensitivity 75% and specificity 68.18%, Figure 2C and Figure 2D, respectively, Table 2). The data are summarized in Table 2.

Since the two variables, P2X7R levels and TNAP activity, were measured for every patient, in the second step, we decided to analyze whether a combined ROC curve (combiROC) using both variables would improve the accuracy in discriminating between MCI and AD patients. The resulting combiROC did not show significantly higher AUC values than those obtained by individual ROC curves (Figure 2E,F and Table 3).

### 2.5. Longitudinal Studies Confirm That Patients Progressing to MCI Condition Experience Changes in Their Plasma TNAP Activity and P2X7 Levels

To evaluate how the progression of cognitive decline could affect both the P2X7R levels and TNAP activity in plasma, we measured both parameters in patients who had been classified as being apparently cognitively unimpaired (CU) at the first medical visit (V1) but who had progressed to MCI 6 years later (in their sixth visit, V6) and compared them with stable control patients. While, in stable control patients, both P2X7R levels and TNAP activity changed insignificantly with time (Figure 3A,B), in the CU patients who progressed to MCI, both TNAP activity and P2X7 levels significantly increased at V6 (Figure 3C,D).

Finally, to determine whether these parameters may be useful in identifying CU older individuals, we compared the two parameters at V1 both in CU individuals and in stable controls. Interestingly, CU individuals presented significantly lower TNAP activity on their first visit than the stable controls (Figure 3F). However, we did not detect any significant modifications in their plasma P2X7 levels before they progressed to MCI (Figure 3E).

## 3. Discussion

In this study, we investigated whether two proteins whose expression is increased in the brains of AD patients, the purinergic receptor P2X7R and the ectoenzyme TNAP [17,18,22,23], are also altered in the plasma of these patients and therefore could potentially be used as plasma biomarkers to predict the evolution of the disease. Using a small but well-characterized cohort of 60 individuals equally distributed into non-affected controls, MCI patients and AD patients, we found that (i) the baseline TNAP plasma activity was increased in MCI patients but not in AD patients, (ii) AD patients but not MCI patients showed higher baseline plasma P2X7R levels than non-affected controls, (iii) the longitudinal analyses, in addition to confirming that those individuals progressing to MCI had increased TNAP activity in plasma, also revealed that CU individuals had lower plasma TNAP activity than stable controls. Hence, our data support that the measurement of TNAP plasma activity has excellent diagnostic accuracy in identifying MCI patients, whereas the determination of plasma P2X7R levels has high diagnostic accuracy in identifying AD patients.

Previous studies reported that AD patients presented a slight (around 12%) elevation in the total AP plasma activity that was inversely correlated with cognitive function [24,25]. In addition, here, we found that MCI patients but not AD patients presented a significant increase (around 30%) in TNAP plasma activity. Although our results seem not to be well matched to previous works, we must take into account that (i) there is no total correlation between plasma AP and TNAP activities. Indeed, it has been estimated that only around 75% of plasma AP activity detected is due to TNAP [24], and (ii) there are some differences in the inclusion criteria for AD vs. MCI groups [24,25]. For example, whereas Vardy et al. included people with an MMSE score less than 27 in the AD group [24], here, we described MCI patients as those who had an MMSE score between 26 and 28 (see Table 1). Bearing the aforementioned factors in mind, it is possible to postulate that the previously observed slight increase detected in plasma AP activity might be due to the inclusion of MCI patients in the AD group. In agreement with this hypothesis, longitudinal studies confirmed that CU, the apparently cognitively unimpaired older individuals who progress to MCI, had increased TNAP plasma activity upon progression to MCI. Intriguingly, CU individuals showed a slight but significant decrease in TNAP plasma activity when compared with stable controls. These results suggest that TNAP may be considered a predictive biomarker for AD. Since it is well known that pTau accumulation in the brain occurs years before the onset of cognitive impairment [32], and considering that it was also reported that acute phosphorylated tau (pTau) protein accumulation caused by a brain injury leads to a significant decrease in total plasma AP activity [33], it could be suggested that the reduction in TNAP plasma activity observed in CU may be linked to the initial brain pTau accumulation. Nevertheless, considering the small size of our cohort, additional studies should be conducted to consolidate these results.

Given that recent studies reported that P2X7R plasma levels are increased in pathologies with an underlying inflammatory condition, such as infections, sepsis, diabetes, temporal lobe epilepsy or COVID-19 [26,27,28,34,35], we also measured P2X7R plasma levels during AD progression. Our population analyses revealed that AD but not MCI or CU individuals presented elevated baseline P2X7R plasma levels. It is noteworthy that the predominant hypothesis suggests that there are two neuroinflammation peaks during AD progression, with the first being associated with an anti-inflammatory response and the second taking place during the progression from MCI to AD, and that they are associated with a proinflammatory switch [36,37]. Because it was reported that an increase in P2X7R expression is associated with proinflammatory phenotypes of human microglia [38], it could be suggested that the increase in microglial P2X7R is linked to the increase in P2X7R plasma levels detected in AD. Contrary to this hypothesis, it was reported that AD patients showed reduced monocyte surface P2X7 expression [39], with the main blood cells expressing P2X7Rs [35,40,41]. Interestingly, the authors indicated that this reduction is caused by environmental changes rather than genetic factors [39]. Since the activation of P2X7R in monocytes may lead to its shedding in microparticles [28], it is reasonable to think that the reduction in P2X7R expression observed on monocyte surfaces in AD patients results from the sustained stimulation of P2X7R caused by the high extracellular ATP associated with the inflammatory state [42]. Furthermore, since, unlike population studies, longitudinal studies showed that those patients who progressed to MCI also had increased P2X7R plasma levels, we cannot exclude other factors from contributing to the raised P2X7R plasma levels. However, further studies should be carried out to determine the exact source of plasma P2X7R protein.

Over the last few years, blood-based biomarkers, mainly including Aβ42/40 and phosphorylated tau at different residues, have been shown to have reasonable diagnostic accuracy in distinguishing AD or MCI patients from non-affected controls [9,10,11,12]. Here, we report, for the first time, two new blood-based biomarkers, (i) TNAP plasma activity with the diagnostic accuracy (AUC = 0.75) to distinguish between MCI patients and non-affected controls and (ii) P2X7R plasma levels with the diagnostic accuracy to discriminate between AD and non-affected controls (AUC = 0.73). A recent study comparing eight plasma Aβ42/40 assays in AD [43] found that half of the assays showed AUC values ranging from 0.71 to 0.69. In contrast, another study did not find significant differences between MCI and CU, showing an AUC value of 0.56 for discrimination between the two groups [44]. Here, we report that P2X7R plasma levels show AUC values ranging from 0.73 to 0.65. Thus, based on the aforementioned results and comparing the slight decrease in Aβ42/40 plasma levels detected in AD patients (10–20%) [9] versus the robust increase in plasma P2X7R levels (around 100%), it is plausible to claim that P2X7R plasma levels are a solid biomarker capable of distinguishing AD patients from non-affected controls. Regarding Tau plasma levels, although the p-Tau181 plasma level (Tau phosphorylated at threonine 181) shows outstanding diagnostic accuracy for differentiating AD dementia from non-AD dementia disorders (AUC = 0.94) [10], it presents a more modest AUC value for pTau181 to separate MCI patients who progressed to AD from those that did not [44]. All of these data indicate that we should not consider a single biomarker for use as a stand-alone test, but we should evaluate it within a clinical context and in combination with others to make an accurate diagnosis of AD or MCI. In this context, our data offer a proof of concept that the combined detection of both P2X7R plasma levels and TNAP plasma activity alone or jointly with other biomarkers constitute a promising novel diagnostic tool able to support the differential diagnosis of AD and MCI patients. Moreover, our preliminary results also suggest that the measurement of TNAP plasma activity may be considered a promising prognostic biomarker of MCI. Nevertheless, considering the limited size of the cohort analyzed, the results reported here should be confirmed by subsequent works analyzing larger study cohorts.

## 4. Materials and Methods

### 4.1. Human Subjects/Participants

Elderly cognitively healthy controls and subjects with mild cognitive impairment were selected from the Vallecas Project Cohort, a single-center longitudinal study. The participants of the Vallecas Project (recruited by advertisement) were community-dwelling individuals between 70 and 85 years at baseline, independent in activities of daily living and without any neurological or psychiatric disorders [45]. All participants provided written informed consent to the experimental procedure, and the project was approved by the Ethics Committee of the Instituto de Salud Carlos III.

### 4.2. Neurological and Neuropsychological Assessments

The classification of patients into the control or MCI group was given by the consensus of a neurologist and neuropsychologist and based on neurological and neuropsychological assessments. The neuropsychological assessment battery was performed by neuropsychologists and included, among others, the total scores of the Mini-Mental State Examination (MMSE), the Functional Activities Questionnaire (FAQ) and the Clock Drawing Test (CDT), as well as the Global Depression Scale (GDS) and the Clinical Dementia Rating (CDR), as described by Olazarán et al. (2015) [45]. Participants with MCI were defined using criteria described by Petersen et al. (1999) [46]. In particular, the Mini-Mental Statement Examination (MMSE) followed the practical method proposed by Folstein et al. (1975) [47], and the Functional Activities Questionnaire (FAQ) was performed as reported by Pfeffer et al. (1982) [48]. 

The AD group consisted of clinically diagnosed patients with moderate to severe AD who were institutionalized at the Queen Sofia Foundation Alzheimer Center. The AD-type dementia diagnosis was established according to the National Institute on Neurological Disorders and Stroke and the Alzheimer’s Disease and Related Disorders Association (NINCDS-ADRDA) guidelines [6]. The application of many standard cognitive scales, including the MMSE in moderate to severe AD patients, leads to a well-known floor effect due to the advanced cognitive impairment in this population [49,50]. The MMSE test requires a lot of time to be completed, often leading to fatigue and frustration in patients with advanced AD, and consequently, it severely affects the subject’s performance. A brief (less than 5 min) and Folstein-based MMSE cognitive assessment instrument known as the Severe Mini-Mental State Examination (SMMSE) was used to confirm the classification of subjects into the AD groups [51]. This SMMSE test is useful in the lower range of the MMSE (patients scoring less than 10 on the MMSE). Unfortunately, scoring in both scales gives a range of values between 0 and 30, which can be confusing, and scores from both scales should not be directly compared [52].To avoid misleading readers, we do not include the SMMSE data in Table 1.

### 4.3. Neuroimaging

Participants underwent brain MRI scans with a 3.0T MRI scanner (Sigma HDxt GEHC, Waukesha, WI, USA) based on a standardized protocol. The Fazekas scale was used to quantify the number of T2 hyperintense white matter lesions, usually attributed to small vessel damage [53].

### 4.4. Human Blood Sample Collection

Blood was collected by venipuncture using BD Vacutainer tubes with sodium citrate following the manufacturer’s instructions. Blood samples were processed within 1 h of procurement, and fractions were then aliquoted and stored at −80 °C until use [45].

### 4.5. APOE Genotyping

Genomic DNA was isolated from peripheral blood following standard procedures [54]. The genotypes of *APOE* gene polymorphisms (rs429358 and rs7412) were determined by Real-Time PCR.

### 4.6. ELISA

Human P2X7R protein levels in plasma were measured using a quantitative ELISA system kit (CUSABIO, Houston, TX, USA), as reported previously [26,28]. Plasma samples in a ½ dilution and standards were prepared using sample diluent (final volume, 100 μL), added to 24- or 96-well plates and incubated for 2 h at 37 °C. After the indicated time, the liquid was removed, without washing, and 100 μL of Biotin antibody was added to each well. Incubation was performed for 1 h at 37 °C. The wells were then aspirated and washed three times with wash Buffer (200 μL). After washing, HRP-avidin (100 μL) was added to each well, and again, plates were incubated for 1 h at 37 °C. Then HRP-avidin was removed, and the washing step was repeated five times. After this step, 90 μL of TMB substrate was added to each well and incubated for 15–30 min at 37 °C in the dark. A Stop solution (50 μL) was added to each well, and the optical density was read within 5 min using a microplate reader set to 450 nm. A wavelength correction was performed by subtracting the readings obtained at 540nm due to optical imperfections in the plate. For each standard and sample measurement, the blank absorbance was subtracted. A standard curve was constructed from serial dilutions of the P2X7R Standard that was provided with the commercial kit. The standard curve was constructed by using “Curve Expert” software (https://www.cusabio.com/c-18069.html, accessed on 27 June 2023) and fitting the data to an exponential association model with an r > 0.99. Standards and sample concentrations were calculated. For plasma samples, concentration values were multiplied by the dilution factor.

### 4.7. TNAP Activity

Plasma samples (10 μL) were brought to 0.2 mL with the following reaction conditions: 0.2 M Diethanolamine and 1 mM MgCl (Merck Life Science S.L.U, Madrid, Spain), pH 9.8, incubated with 4 mM p-nitrophenyl phosphate (Merck Life Science S.L.U, Madrid, Spain) at 37 °C with agitation. All samples were assayed in duplicate. The background signal was determined by pre-incubating plasma samples with 150 μM levamisole (a TNAP-specific inhibitor) for 5 min, 37 °C, before adding 4 mM p-nitrophenyl phosphate in the presence of levamisole. The absorbance of the samples was then measured at 405 nm, 25 °C, in a spectrophotometer.

### 4.8. Statistical Analysis

Statistical analysis of the data was performed using Prism 9 (GraphPad, Dotmatics, Houston, TX, USA) and SPSS. Outliers were identified using Prism and excluded from the analysis. Data are mean ± standard error of the mean (SEM). The Shapiro–Wilk test was performed to determine if the data were normally distributed. For non-parametric analysis of two-group comparisons, data were analyzed with the Mann–Whitney test. For parametric analysis, data were analyzed with a two-tailed unpaired Student’s *t*-test for two-group comparisons and with one-way ANOVA followed by Tukey’s post hoc test for multiple comparisons. The statistical test used and *p*-values are indicated in each figure legend. Significance was considered at * *p* < 0.05, ** *p* < 0.01, *** *p* < 0.001 or **** *p* < 0.0001 throughout the study.

Receiver operating characteristic (ROC) analysis was performed to investigate the diagnostic potential of measuring P2X7R protein and TNAP activity changes. Because ROC curves depend on the patients’ characteristics and the disease spectrum, we tested separately for MCI and AD conditions. Individual ROC curves were obtained using Prims 9 and SPSS. ROC curves above the diagonal line are considered to have a reasonable discriminating ability to diagnose patients with and without the disease/condition [55]. The area under the ROC curve (AUC) summarizes the overall diagnostic accuracy of a test. AUC data are shown as AUC value ± standard error (SE). A value of 0 indicates a perfectly inaccurate test, and a value of 1 reflects a perfectly accurate test (AUC = 0.5, no discrimination; AUC = 0.7–0.8, acceptable; AUC = 0.8–0.9, excellent; AUC > 0.9, outstanding). The sensitivity and specificity test cut-off values were determined based on the maximum Youden index, that is to say, the value at which the difference between the sensitivity (total positives) and 1-specificity (false positives) is the maximum (Se+Sp-1). As two variables, P2X7R levels and TNAP activity, were measured for every patient, we can calculate the AUC of combined variables by running a binary logistic regression to determine the probability and then obtain the ROC curve using the probability as the test variable. This is known as a combined ROC curve.

## Figures and Tables

**Figure 1 ijms-24-10897-f001:**
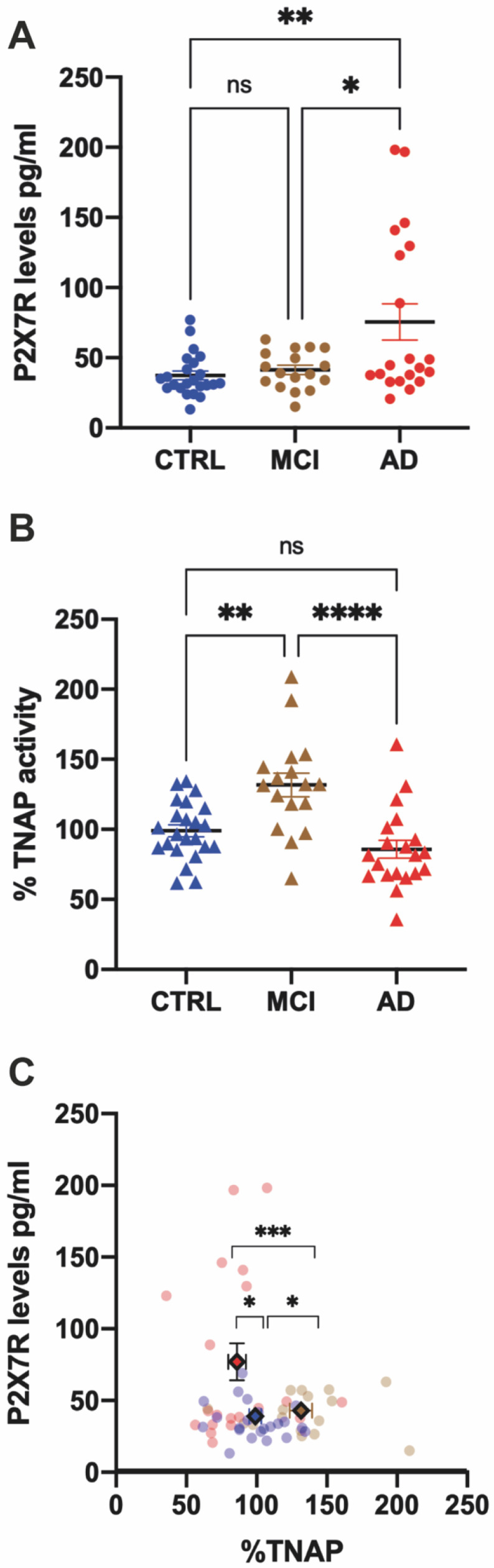
Measurement of P2X7R levels and TNAP activity in the plasma samples of the same cohort of control, MCI and AD patients. (**A**) P2X7 receptor levels in plasma samples from Control (*n* = 23, blue), MCI (*n* = 17, brown) and AD (*n* = 20, red) patients. Plasma samples were analyzed via P2X7R-detecting ELISA. Data are presented as means ± SEM. (**B**) TNAP activity in the plasma of the same cohort of patients. Data are given as a percentage relative to the control, and they are represented as means ± SEM. Significance measured by ordinary one-way ANOVA with Tukey’s multiple comparisons test. **** *p* < 0.0001, ** *p* < 0.01, * *p* < 0.05. (**C**) Scatter plot of both P2X7R and TNAP activity for each individual patient (colored circles). Each color encodes a specific group: blue—control; brown—MCI; and red—AD. The center of mass for each experimental group is represented by a colored diamond, whose coordinates are the average of P2X7R levels and TNAP activity (mean ± SEM). Hotelling’s T^2^ test indicates that the centers of mass for the groups are significantly different from one another. *** *p* < 0.001, * *p* < 0.05.

**Figure 2 ijms-24-10897-f002:**
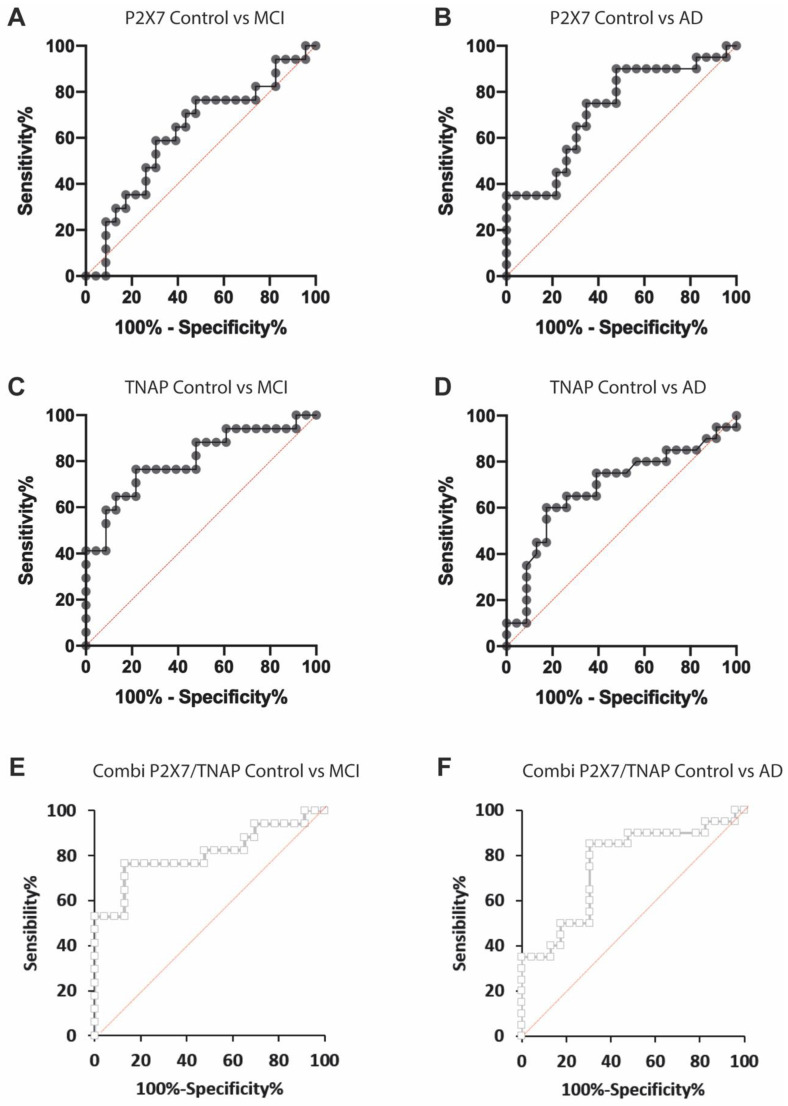
Receiver operating characteristic (ROC) analysis to investigate the diagnostic potential of measuring plasma P2X7R protein and TNAP activity. The area under the curve (AUC) summarizes the overall diagnostic accuracy of the test (AUC = 0.5, no discrimination; AUC = 0.7–0.8, acceptable; AUC = 0.8–0.9, excellent; AUC > 0.9, outstanding). The sensitivity and the sensitivity test cut-off values were determined based on the maximum Youden index. (**A**,**B**) Individual ROC curves (•) to test the diagnostic potential of measuring P2X7R levels for MCI and AD, respectively. (**C**,**D**) Individual ROC curves (•) to test the diagnostic potential of measuring TNAP activity for MCI and AD, respectively. CombiROC (□) predicts the accuracy in differentiating between control and MCI (**E**) and between control and AD (**F**) patients by measuring both P2X7R and TNAP activity.

**Figure 3 ijms-24-10897-f003:**
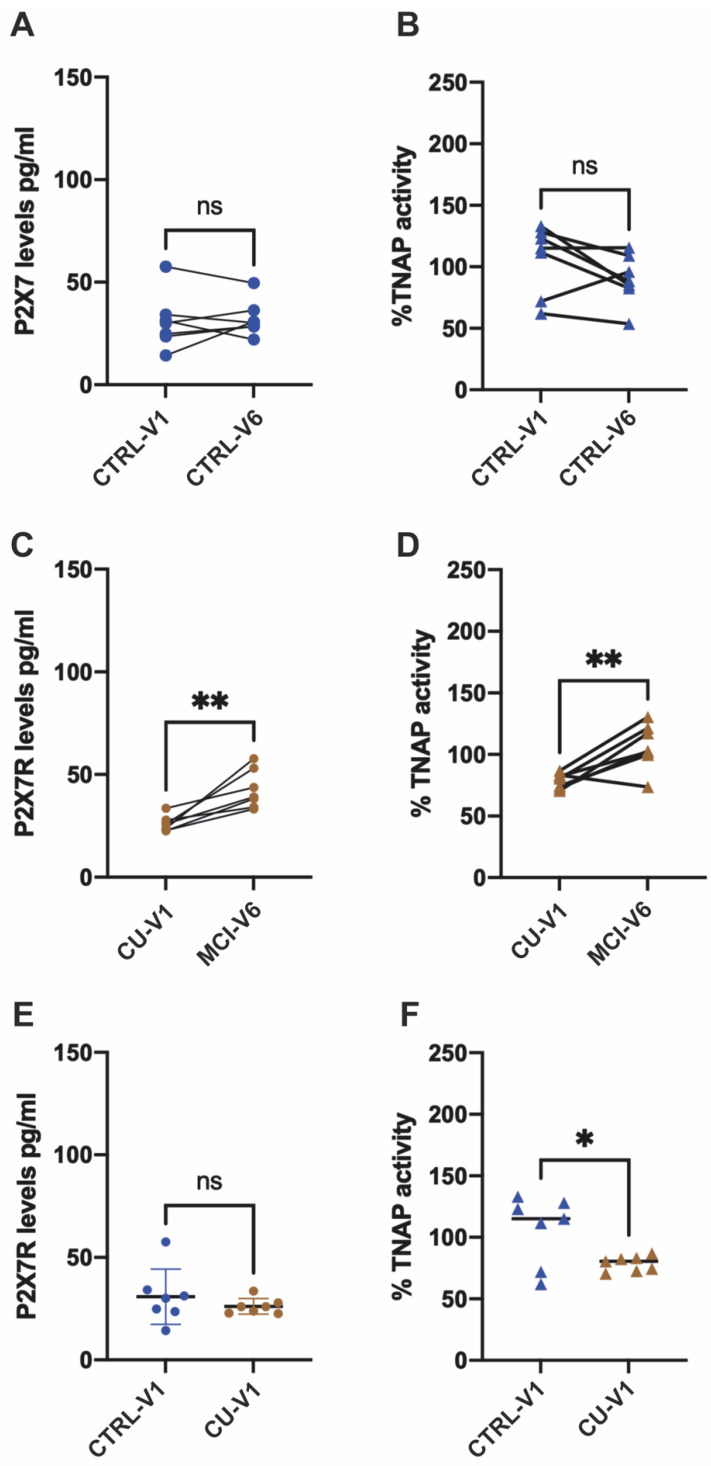
Longitudinal study of stable control patients and MCI converter patients. P2X7R levels and TNAP activity in patients who were classified as being apparently cognitively unimpaired (CU) at the first medical visit but who had progressed to MCI 6 years later (CU-V1 and MCI-V6, *n* = 7, brown) compared with those in 7 control stable patients (CTRL-V1 and CTRL-V6, *n* = 7, blue). (**A**–**D**) Dot plot showing P2X7R plasma levels (**A**,**C**) and TNAP activity (**B**,**D**) in the same patients at visit 1 and visit 6. Out of 7 stable control patients, there were no differences in P2X7R plasma levels when compared to V1 (**A**,**B**). In contrast, 7 patients who progressed to MCI showed significant differences in P2X7R levels (**C**,**D**). (**E**) Comparison of P2X7R levels between stable control group and CU at visit 1. (**F**) Comparison of %TNAP activity between the stable control group and CU at visit 1. Significance measured by parametric paired and unpaired *t*-tests. ns: not significant, ** *p* < 0.01, * *p* < 0.05.

**Table 1 ijms-24-10897-t001:** Summary of demographic and basic clinical data for all groups.

	Control *n* = 23		MCI *n* = 17		AD *n* = 20		*p*-Value
	Mean	SEM	Mean	SEM	Mean	SEM	
Age (years)	77.0	0.5	79.9	0.7	78.6	1.3	*P*_ANOVA_ = 0.09 (ns)
Sex F/M %	60/40		41/59		55/45		
Fazekas scale value	0.8	0.2	1.2	0.2	1.6	0.2	*P*_ANOVA_ = 0.045 (*)
%FS = 0	35		29		13		
%FS = 1	52		29		38		
%FS = 2	9		35		31		
%FS = 3	4		6		19		
MMSE	29.1	1.1	27.3	1.4	-	-	*P*_MW_ < 0.0001
FAQ	0.6	0.9	3.0	1.8	-	-	*P*_MW_ < 0.0001
CDT	9.5	1.0	8.5	1.7	-	-	*P*_MW_ = 0.0171
GDS	1.0	1.4	1.8	2.1	-	-	*P*_MW_ = 0.106
CDR	0.0	0.0	0.5	0.0	-	-	*P*_MW_ < 0.0001
*APOE_Ɛ_3*_Ɛ_4 allele, %	8.7		23.5		40		
*APOE_Ɛ_4*_Ɛ_4 allele, %	0		0		15		

NOTE. Mini-Mental State Examination (MMSE) is a global cognitive test. A diagnosis of cognitive decline is made when a score below 24 is achieved. The Functional Activities Questionnaire (FAQ) measures the loss of patient autonomy in activities of daily living. A diagnosis of AD is made when the FAQ score is above 6. The Clock Drawing Test (CDT) is widely used as a visual-spatial ability test and screening test for dementia patients. The results range from 10 (best) to 1 (worst), and a performance score below 6 indicates a dementia condition. The GDS (global depression scale) cut point is greater than 5. The Clinical Dementia Rating (CDR) is based on a scale of 0–3 (o = no dementia; 0.5 = questionable dementia; 1 = MCI; 2 = mild cognitive impairment, 3 = severe cognitive impairment). MCI, mild cognitive impairment; AD, Alzheimer’s disease; SEM, standard error mean; ANOVA, analysis of variance; MW, Mann–Whitney test; * *p* < 0.05; F, female; M, male.

**Table 2 ijms-24-10897-t002:** Summary of individual ROC curve data for all groups.

		AUC (±SE)	IC95%	Cut-Off Value	Sensitivity %	Specificity %	*p*-Value	
**CTRL vs. MCI**	P2X7R pg/mL	0.65 ± 0.09	0.46–0.83	>32.5	78.6	50.0	0.144	
	TNAP activity	0.75 ± 0.09	0.58–0.92	>112.6	71.4	72.3	0.011	* *p* < 0.05
**CTRL vs. AD**	P2X7R pg/mL	0.73 ± 0.08	0.58–0.89	>32.3	90.00	50.22	0.009	** *p* < 0.01
	TNAP activity	0.69 ± 0.08	0.53–0.86	<92.9	75.00	68.18	0.029	* *p* < 0.05

**Table 3 ijms-24-10897-t003:** Summary of combiROC curve data.

		AUC (±SE)	IC95%	Cut-Off Predicted Probability	Sensitivity %	Specificity %
**CTRL vs. MCI**	Probability of the combined variable	0.76 ± 0.09	0.59–0.94	<0.49	71.4	86.4
**CTRL vs. AD**	Probability of the combined variable	0.77 ± 0.07	0.63–0.92	<0.40	80.00	72.7

## Data Availability

The data that support the findings of this study are available from the corresponding author, MD-H, upon reasonable request.

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
