# Peer review of "TNAP and P2X7R: New Plasma Biomarkers for Alzheimer’s Disease"

_ijms, 2023, doi:10.3390/ijms241310897_

Round 1

Reviewer 1 Report

In this paper TNAP and P2X7R plasma levels are assessed as a biomarkers for Alzheimer disease.

In material and methods, in 4.2. section: It is not well explained how the cognitive status of the subjects was determined. The reference by Olazarán (29) does not provide the information. This data is very important because it is the basis for the discussion, since the scores used are not the same and cause confusion in the arguments (line 247 and following)

In the discussion, they use CU and MCI interchangeably, it is the same (line 256, 261).

On line 242, they say, however, here we found... in my opinion it's better to say, besides.

What was APOE genotyped for?

Author Response

.- In this paper TNAP and P2X7R plasma levels are assessed as a biomarkers for Alzheimer disease.

Firstly, we want to thank the reviewer for her/his kindly comments and interesting suggestions. Considering his/her comments we have modified the text and we think that helps to clarify his/her major concerns and also contribute to the strengthen and refine the main conclusions of our work.

.- In material and methods, in 4.2. section: It is not well explained how the cognitive status of the subjects was determined. The reference by Olazarán (29) does not provide the information. This data is very important because it is the basis for the discussion, since the scores used are not the same and cause confusion in the arguments (line 247 and following).

We thank the reviewer for her/his comment. In order to clarify this, we have rephrased the 4.2 section as follows: page 12, line 331; “Classification of patients into Control or MCI group was given by consensus of neurologist and neuropsychologist and based on the neurological and neuropsychological assessment. The neuropsychological assessment battery was performed by neuropsychologists and include, among others, the total scores of the Mini Mental State Examination (MMSE), the Functional Activities Questionnaire (FAQ) and the Clock Drawing Test (CDT), as well as the Global depression Scale (GDS) and the Clinical Dementia Rating (CDR), as described in Olazarán et al 2015. Participants with MCI were defined using criteria described by Petersen et al., 1999 [45]. In particular, the Mini Mental Statement Examination (MMSE) followed the practical method proposed by Folstein et al. 1975, [46] and the Functional Activities Questionnaire (FAQ) was performed as in Pfeffer et al. 1982 [47]

The AD group consists of clinically diagnosed patients with moderate to severe AD patients that were institutionalized at the Queen Sofia Foundation Alzheimer Center. AD-type dementia diagnosis was established according to the National Institute on Neurological Disorders and Stroke, and the Alzheimer's Disease and Related Disorders Association (NINCDS-ADRDA) guidelines [6]. Application of many standard cognitive scales, including MMSE in moderate to severe AD patients, leads to a well-known floor effect due to the advanced cognitive impairment in this population [48, 49]. The MMSE test requires a lot of time to be completed, often leading to fatigue and frustration in patients with advanced AD, and consequently, it severely affects the subject´s performance. A brief (less than 5 minutes) and Folstein-based MMSE cognitive assessment instrument known as Severe Mini-Mental State Examination (SMMSE) was used to confirm the classification of subjects into the AD groups [50]. This SMMSE test is useful in the lower range of the MMSE (patients scoring less than 10 on the MMSE). Unfortunately, scoring in both scales gives a range of values between 0 and 30 that can be confusing, and scores from both scales should not be directly compared [51].To avoid misleading, we do not include the SMMSE data in Table 1.”

.- In the discussion, they use CU and MCI interchangeably, it is the same (line 256, 261).

We appreciate the reviewer´s comments, and to clarify this point we have rewritten the indicated sentences as follows, page 10, line 257 “Agree with this hypothesis, longitudinal studies confirmed that CU, the no apparent cognitive unimpaired older individuals who are going to progress to MCI increased their TNAP plasma activity upon conversion to MCI. Intriguingly, CU individuals showed a slight, but significant decrease of TNAP plasma activity when compared with stable controls. These results are suggesting that TNAP may be considered as a predictive biomarker for AD.”

.- On line 242, they say, however, here we found... in my opinion it's better to say, besides.

We want to thank the reviewers for the editing work done. Following his/her suggestions we added the suggested change in the text page 10 line 246.

What was APOE genotyped for?

We appreciate the interesting reviewer´s question. As he/she is well-known, the APOE Ɛ4 allele is the strongest risk factor for late-onset AD and is increased up to ~40% in patients with AD. Despite our limited AD cohort size, we wanted to confirm that it preserves both the expected Fazekas scale and the frequency of the APOE Ɛ4 allele described in wider populations.

Reviewer 2 Report

This study aimed to identify new blood biomarkers for Alzheimer´s disease (AD) for early diagnosis. The authors have tested purinergic receptor P2X7 and the tissue non-specific alkaline phosphatase ectoenzyme (TNAP) and P2X7R in blood samples from AD, mild cognitive impairment, and control patients.

They found AD but not MCI patients present increased plasma P2X7R levels and that TNAP plasma activity was increased in MCI and decreased in the AD group.

Despite the low number of participants in the study, the study is relevant and I recommend it for publication. However, a minor revision can be performed.

1. To correct for moderate to mild cognitive impairment across the entire manuscript.

2. In the methods, it could be better explained how patients were classified into control, MCI, or AD groups.

3. In conclusion, the authors can make it it clear that more studies with more participants are needed.

This study aimed to identify new blood biomarkers for Alzheimer´s disease (AD) for early diagnosis. The authors have tested purinergic receptor P2X7 and the tissue non-specific alkaline phosphatase ectoenzyme (TNAP) and P2X7R in blood samples from AD, mild cognitive impairment, and control patients.

They found AD but not MCI patients present increased plasma P2X7R levels and that TNAP plasma activity was increased in MCI and decreased in the AD group.

Despite the low number of participants in the study, the study is relevant and I recommend it for publication. However, a minor revision can be performed.

1. To correct for moderate to mild cognitive impairment across the entire manuscript.

2. In the methods, it could be better explained how patients were classified into control, MCI, or AD groups.

3. In conclusion, the authors can make it clearer that more studies with more participants are needed.

Author Response

This study aimed to identify new blood biomarkers for Alzheimer´s disease (AD) for early diagnosis. The authors have tested purinergic receptor P2X7 and the tissue non-specific alkaline phosphatase ectoenzyme (TNAP) and P2X7R in blood samples from AD, mild cognitive impairment, and control patients.

They found AD but not MCI patients present increased plasma P2X7R levels and that TNAP plasma activity was increased in MCI and decreased in the AD group.

Despite the low number of participants in the study, the study is relevant and I recommend it for publication. However, a minor revision can be performed.

Firstly, we want to thank the reviewer for her/his kindly comments and interesting suggestions. Considering his/her comments we have performed a battery of changes in the text that we think help to clarify his/her major concerns and also contribute to the strengthen and refine the main conclusions of our work.

  1. To correct for moderate to mild cognitive impairment across the entire manuscript.

We want to thank the reviewer for his/her editing work. Following his/her suggestions we have changed moderate per mild across the entire manuscript.

  1. In the methods, it could be better explained how patients were classified into control, MCI, or AD groups.

We thank the reviewer for her/his comment. In order to clarify this, we have rephrased the 4.2 section as follows: page 12, line 331; “Classification of patients into Control or MCI group was given by consensus of neurologist and neuropsychologist and based on the neurological and neuropsychological assessment. The neuropsychological assessment battery was performed by neuropsychologists and include, among others, the total scores of the Mini Mental State Examination (MMSE), the Functional Activities Questionnaire (FAQ) and the Clock Drawing Test (CDT), as well as the Global depression Scale (GDS) and the Clinical Dementia Rating (CDR), as described in Olazarán et al 2015. Participants with MCI were defined using criteria described by Petersen et al., 1999 [45]. In particular, the Mini Mental Statement Examination (MMSE) followed the practical method proposed by Folstein et al. 1975, [46] and the Functional Activities Questionnaire (FAQ) was performed as in Pfeffer et al. 1982 [47]

The AD group consists of clinically diagnosed patients with moderate to severe AD patients that were institutionalized at the Queen Sofia Foundation Alzheimer Center. AD-type dementia diagnosis was established according to the National Institute on Neurological Disorders and Stroke, and the Alzheimer's Disease and Related Disorders Association (NINCDS-ADRDA) guidelines [6]. Application of many standard cognitive scales, including MMSE in moderate to severe AD patients, leads to a well-known floor effect due to the advanced cognitive impairment in this population [48, 49]. The MMSE test requires a lot of time to be completed, often leading to fatigue and frustration in patients with advanced AD, and consequently, it severely affects the subject´s performance. A brief (less than 5 minutes) and Folstein-based MMSE cognitive assessment instrument known as Severe Mini-Mental State Examination (SMMSE) was used to confirm the classification of subjects into the AD groups [50]. This SMMSE test is useful in the lower range of the MMSE (patients scoring less than 10 on the MMSE). Unfortunately, scoring in both scales gives a range of values between 0 and 30 that can be confusing, and scores from both scales should not be directly compared [51].To avoid misleading, we do not include the SMMSE data in Table 1.)”.

  1. In conclusion, the authors can make it it clear that more studies with more participants are needed.

Following the reviewer´s suggestion, we have rewritten the last paragraph of the discussion section to clarify the reviewer-indicated point as follows (page 11, line 318), “Nevertheless, considering the limited size of the cohort analyzed, the results reported here should be confirmed by subsequent works analyzing larger study cohorts.”

Reviewer 3 Report

The article presents an interesting original study exploring new blood biomarkers for identifying the progression of Alzheimer's disease (AD). P2X7R levels and TNAP activity in the plasma are proposed as diagnostic markers. Despite the small cohort size, the study includes longitudinal observations, which are a key element in recording the progressive changes in the disease course. However, the article needs revision and further considerations.

Major comments:

Include correlations between MRI Fazekas scale and P2X7R levels in the plasma.

Include correlations between MRI Fazekas scale and TNAP activity in the plasma.

Similarly, perform correlation analyses for all neurocognitive assessments:

Mini-Mental State Examination score vs. P2X7R levels.

Functional Activities Questionnaire vs. P2X7R levels in the plasma.

Global Depression Scale vs. P2X7R levels in the plasma.

Clock drawing test vs. P2X7R levels in the plasma.

Clinical dementia rating vs. P2X7R levels in the plasma.

Mini-Mental State Examination score vs. TNAP activity in the plasma.

Functional Activities Questionnaire vs. TNAP activity in the plasma.

Global Depression Scale vs. TNAP activity in the plasma.

Clock drawing test vs. TNAP activity in the plasma.

Clinical dementia rating vs. TNAP activity in the plasma.

Provide an explanation for the absence of neurocognitive assessments for AD patients in the study.

Address the discrepancy between the AUC value of 0.80 mentioned in the text and the corresponding value in Table 3.

The naming of "mci-v1" and referring to it as "controls at visit 1" is misleading. Rename and revise this in Figure 4 and its description accordingly.

Properly mention MCI and CU (cognitively unimpaired) in the results section. Provide information on the number of MCI and CU participants out of the total cohort size.

Note that cited (42 ref) study has a cohort population eight times larger than your study. Comparing only the AUC values does not suggest that P2X7R diagnostic accuracy is superior. Consider revising the following phrase: "Remarkably, a recent study comparing eight plasma Aβ42/40 assays in AD (42) found that half of the assays showed AUC values (ranging from 0.71 to 0.69) equal to or lower than those detected in our study by measuring P2X7R plasma levels." This phrase is misleading.

Minor comments

Please correct the typo in the author's name: "Miguel Diaz-Hernandez" instead of "Miguel Diaz-Hernandez and 1,3,*."

It is noted that Figure 1 does not provide any new information as the data is already presented in Table 1. Consider removing the figure.

In line 105, replace "clinical dementia rating score total" with "clinical dementia rating."

In line 112, change "bellow" to "below."

In line 114, replace "rate" with "rating."

In lines 291 to 293, avoid using terms like "excellent" and "high” diagnostic accuracy.

Ensure consistency in the notation of wavelength values (e.g., 450 nm and 540 nm).

Ensure proper capitalization in several instances, such as "blue-control," "brown-MCI," and "red-AD" in Figure 2.

Author Response

.- The article presents an interesting original study exploring new blood biomarkers for identifying the progression of Alzheimer's disease (AD). P2X7R levels and TNAP activity in the plasma are proposed as diagnostic markers. Despite the small cohort size, the study includes longitudinal observations, which are a key element in recording the progressive changes in the disease course. However, the article needs revision and further considerations.

Firstly, we want to thank the reviewer for her/his kindly comments and interesting suggestions. Considering his/her comments we have modified the text and figures that we think help to clarify his/her major concerns and also contribute to the strengthen and refine the main conclusions of our work.

Major comments:

.- Include correlations between MRI Fazekas scale and P2X7R levels in the plasma.

Include correlations between MRI Fazekas scale and TNAP activity in the plasma.

Similarly, perform correlation analyses for all neurocognitive assessments:

Mini-Mental State Examination score vs. P2X7R levels.

Functional Activities Questionnaire vs. P2X7R levels in the plasma.

Global Depression Scale vs. P2X7R levels in the plasma.

Clock drawing test vs. P2X7R levels in the plasma.

Clinical dementia rating vs. P2X7R levels in the plasma.

Mini-Mental State Examination score vs. TNAP activity in the plasma.

Functional Activities Questionnaire vs. TNAP activity in the plasma.

Global Depression Scale vs. TNAP activity in the plasma.

Clock drawing test vs. TNAP activity in the plasma.

Clinical dementia rating vs. TNAP activity in the plasma.

We want to thank the reviewer´s suggestion. To address this request, we made all the suggested correlation analyses and have annexed the data to the paper (Anexo 1 and 2), except for the correlation with the Clinical Dementia Rating (CDR). As CDR is a binary score (Control=0; MCI=0,5), the correlation analysis with the two parameters gives rise to a meaningless horizontal line.

We have modified the text and included the results of the analysis in page 5, line 162 as follows: “Correlation analysis between MRI Fazekas scale and P2X7R levels or TNAP activity in all diagnostic groups was non-statistically significant (see Anexo 1). In the same line, we performed the correlation analysis between all the neuropsychological assessments and both parameters for Control and MCI groups (Anexo 2). P2X7R levels reached only a significant positive correlation with MMSE in the MCI group. Again, only in the MCI group, a significant positive correlation was obtained between TNAP activity and the Global Depression Scale, consistent with recently published data [31].”

Provide an explanation for the absence of neurocognitive assessments for AD patients in the study.

We agree with the reviewer´s concern and made it clearer in the material and method section by adding the following paragraph, page 12, line 331: “Classification of patients into Control or MCI group was given by consensus of neurologist and neuropsychologist and based on the neurological and neuropsychological assessment. The neuropsychological assessment battery was performed by neuropsychologists and include, among others, the total scores of the Mini Mental State Examination (MMSE), the Functional Activities Questionnaire (FAQ) and the Clock Drawing Test (CDT), as well as the Global depression Scale (GDS) and the Clinical Dementia Rating (CDR), as described in Olazarán et al 2015. Participants with MCI were defined using criteria described by Petersen et al., 1999 [45]. In particular, the Mini Mental Statement Examination (MMSE) followed the practical method proposed by Folstein et al. 1975, [46] and the Functional Activities Questionnaire (FAQ) was performed as in Pfeffer et al. 1982 [47]

The AD group consists of clinically diagnosed patients with moderate to severe AD patients that were institutionalized at the Queen Sofia Foundation Alzheimer Center. AD-type dementia diagnosis was established according to the National Institute on Neurological Disorders and Stroke, and the Alzheimer's Disease and Related Disorders Association (NINCDS-ADRDA) guidelines [6]. Application of many standard cognitive scales, including MMSE in moderate to severe AD patients, leads to a well-known floor effect due to the advanced cognitive impairment in this population [48, 49]. The MMSE test requires a lot of time to be completed, often leading to fatigue and frustration in patients with advanced AD, and consequently, it severely affects the subject´s performance. A brief (less than 5 minutes) and Folstein-based MMSE cognitive assessment instrument known as Severe Mini-Mental State Examination (SMMSE) was used to confirm the classification of subjects into the AD groups [50]. This SMMSE test is useful in the lower range of the MMSE (patients scoring less than 10 on the MMSE). Unfortunately, scoring in both scales gives a range of values between 0 and 30 that can be confusing, and scores from both scales should not be directly compared [51].To avoid misleading, we do not include the SMMSE data in Table 1.”  

.- Address the discrepancy between the AUC value of 0.80 mentioned in the text and the corresponding value in Table 3.

We want to thank the reviewer for catching this typo. It has been appropriately amended in the new manuscript´s version (see page 11, line 296).

.- The naming of "mci-v1" and referring to it as "controls at visit 1" is misleading. Rename and revise this in Figure 4 and its description accordingly.

The reviewer is right, and following his/her suggestion, we have modified the name of this group as is now indicated in both Figure 3 and text page 8, line 206 as follows; “To evaluate how the progress of cognitive decline could affect both P2X7R levels and the TNAP activity in plasma, we measured both parameters in patients that had been classified as no apparent cognitive unimpaired (CU) at the first medical visit (V1), but that converted into MCI 6 years later (in their sixth visit, V6), and compared them with stable Control patients.”

.- Properly mention MCI and CU (cognitively unimpaired) in the results section. Provide information on the number of MCI and CU participants out of the total cohort size.

This issue has already been addressed in the previous point. The number of all individuals analyzed per group was summarized in Table 1. As it is explained, CU were those individuals classified as no-apparent cognitive unimpaired at the first medical visit but converted into MCI 6 years later at visit 6.

.- Note that cited (42 ref) study has a cohort population eight times larger than your study. Comparing only the AUC values does not suggest that P2X7R diagnostic accuracy is superior. Consider revising the following phrase: "Remarkably, a recent study comparing eight plasma Aβ42/40 assays in AD (42) found that half of the assays showed AUC values (ranging from 0.71 to 0.69) equal to or lower than those detected in our study by measuring P2X7R plasma levels." This phrase is misleading.

We thank the reviewer for her/his interesting comment. Following his/her suggestion we have edited the text as follows page 11, line 298 “A recent study comparing eight plasma Aβ42/40 assays in AD (42) found that half of the assays showed AUC values ranging from 0.71 to 0.69.” And we have incorporated a phrase on line 302 “Here we present that P2X7R plasma levels show AUC values ranging from 0.73 to 0.65. Thus, based …”.

Minor comments:

We want to thank the reviewer for the editing work that improved the manuscript. We have included all the suggested changes by him/her as described below.  

.- Please correct the typo in the author's name: "Miguel Diaz-Hernandez" instead of "Miguel Diaz-Hernandez and 1,3,*." 

The suggested change was added on page 1, line 5

.- It is noted that Figure 1 does not provide any new information as the data is already presented in Table 1. Consider removing the figure.

The reviewer is right, and following his/her suggestion, we removed Figure 1 from the manuscript, which led to renumbering the manuscript´s figures.

.- In line 105, replace "clinical dementia rating score total" with "clinical dementia rating."

The suggested change was added on page 3, line 109

.- In line 112, change "bellow" to "below." Done

The suggested change was added on page 3, line 120

.- In line 114, replace "rate" with "rating."

The suggested change was added on page 3, line 122

.- In lines 291 to 293, avoid using terms like "excellent" and "high” diagnostic accuracy.

Following the reviewer´s suggestion, we removed both words from the text, page 10 lines 296 and 297.

.- Ensure consistency in the notation of wavelength values (e.g., 450 nm and 540 nm).

To clary the reviewer´s concern, let us explain that we have performed two different wavelength measurements in order to measure P2X7R levels more accurately. The manufacturer protocol specified the following instruction “Subtract readings at 540 nm or 570 nm from the readings at 450 nm. This subtraction will correct for optical imperfections in the plate. Readings made directly at 450 nm without correction may be higher and less accurate.”

.- Ensure proper capitalization in several instances, such as "blue-control," "brown-MCI," and "red-AD" in Figure 2.

Again, we thank the reviewer for the editing work. We revised and actualized the capitalization of several words throughout the manuscript.

Round 2

Reviewer 3 Report

The authors have made significant improvements to the manuscript and have conducted additional correlation assays. While the article could benefit from further enhancements in terms of its design, I do not have any major comments on the present version. Therefore, I recommend this paper to the handling editor.